# Assessment of oxidative stress in autism spectrum disorder using reactive oxygen metabolites and biological antioxidant potential

**Masahito Morimoto**[1]*, **Toshiaki Hashimoto**[2◉], **Yoshimi Tsuda**[2‡], **Tadanori Nakatsu**[2‡], **Taisuke Kitaoka**[3◉], **Shojiro Kyotani**[3◉]

**1** Department of pharmacy, Japanese Red Cross Tokushima Hinomine Rehabilitation Center for People with Disabilities, Tokushima, Japan, **2** Department of pediatrics, Japanese Red Cross Tokushima Hinomine Rehabilitation Center for People with Disabilities, Tokushima, Japan, **3** Graduate School of Pharmaceutical Sciences, Tokushima Bunri University, Tokushima, Japan

◉ These authors contributed equally to this work.
‡ These authors also contributed equally to this work.
* morimoto@hinomine-mrc.jp

**Data Availability Statement:** All relevant data are within the paper and its Supporting Information files.

## Abstract

There are several studies on oxidative stress of Autism Spectrum Disorder (ASD), but in these cases there is no study to measure oxidative stress and antioxidant capacity at the same time or studies considering childhood development. Therefore, this study comprehensively assessed the level of oxidative stress in ASD children by simultaneously measuring reactive oxygen metabolites (d-ROMs) and biological antioxidant potential (BAP). The subjects were Japanese, 77 typical development (TD) children, 98 ASD children, samples were plasma. The subjects were divided into age groups: toddlers/preschool age (2–6 years) and school age (7–15 years), to compare the relationships among the d-ROMs levels and BAP/ d-ROMs ratios. Furthermore, the correlations between the Parent-interview ASD Rating Scales (PARS) scores and the measured values were analyzed. The levels of d-ROMs were significantly higher in the ASD (7–15 years) than in TD (7–15 years). The PARS scores were significantly higher in the ASD and were significantly correlated with d-ROMs levels. These results suggested that d-ROMs and BAP/d-ROMs ratios could be objective, measured indicators that could be used in clinical practice to assess stress in ASD children.

## Introduction

Autism spectrum disorder (ASD) is a neurodevelopmental disorder that is characterized by the core symptoms of lack of social communication skills and restricted behaviors and interests, and its prevalence is on the increase [1]. Some children with ASD have secondary symptoms caused by the accumulation of psychological stress, such as insomnia, depression and anxiety [2, 3], and a high proportion of them show aggressive behaviors, self-mutilation behaviors and pica [4]. On the other hand, factors associated with encephalopathy in ASD, such as

**Funding:** The authors received no specific funding for this work.

**Competing interests:** The authors have declared that no competing interests exist.

autoimmune disorders, inflammatory changes in the brain and mitochondrial dysfunction, have been reported, and oxidative stress is assumed to be enhanced in these conditions [5–7].

At present, ASD is diagnosed and evaluated according to the Diagnostic and Statistical Manual of Mental Disorders, Fifth edition (DSM-5) criteria, but there are no objective/biological test criteria. Therefore, recently, various biomarkers have been explored. In particular, many oxidative stress markers have been reported [8–11], including nitric oxide synthase (NOS), xanthine oxidase (XO), glutathione S-transferase (GST), paraoxonase-1 (PON-1), glutathione, methionine and cysteine, which have been suggested to be associated with neurodevelopmental disorders and oxidative stress [9, 10]. However, these markers cannot be measured quickly in clinical settings because of the need for specialized test equipment, or cannot be used comprehensively to assess stress in vivo.

As with other conditions, it is beneficial for patients that tests which affect the assessment of ASD are quick and easy. Therefore, we paid attention to oxidative stress markers, namely, reactive oxygen metabolites (d-ROMs) and biological antioxidant potential (BAP). There has been no report on the relationship between ASD and d-ROMs test. However, the d-ROMs test is a method in which serum hydroperoxides generates alkoxy and peroxy radicals by the Fenton reaction with iron ions, and quantitatively measures them. There are several reports on the relationship between oxidative stress evaluation by ASD and Fenton reaction [12, 13], and d-ROMs seems to be a method applicable to oxidative stress evaluation as well. The BAP test is a test for examining the ability to reduce ferric ions ($Fe3^+$) to ferrous ions ($Fe2^+$). There are reports investigating the relationship between inflammation markers and iron reduction in ASD, suggesting that it can be applied to the evaluation of oxidative stress in ASD [14]. These can be measured quickly and efficiently in about 5 minutes for each test [15] and are used as assessment indicators in various diseases [16–18]. The purpose of this study was to investigate whether d-ROMs and BAP could be objective indicators to assess oxidative stress in untreated ASD children.

## Methods

### Subjects

The subjects were Japanese children aged 2 to 15 years, including 77 typical development (TD) children (39 males and 38 females) and 98 children with untreated ASD (73 males and 25 females). "TD" children included children who were medically judged to be mentally and physically healthy by pediatric specialists; those who 1) had underlying disease, 2) were receiving therapeutic or prophylactic medication, and 3) had suffered from any disease within the previous one month were excluded from the healthy group of children. The "ASD" children included children who had been medically diagnosed by pediatric neurologists as having ASD based on the DSM-5 criteria and clinical symptoms; children who 1) had underlying disease other than the developmental disorders, 2) had received therapeutic intervention for ASD, and 3) were receiving any drugs were excluded from the ASD group of children. As comorbidities, insomnia 20(20.4%), depression 8(8.2%) and anxiety 37(37.8%) were observed.

In our previous study, younger children had higher oxidative stress levels [19]. And it has been reported that the stress level of children differs due to changes in the environment between preschool and school life [20]. Since age and living environment may affect stress, both TD and ASD children were divided into two groups, toddlers/preschool age children (aged 2 to 6 years) and school age children (aged 7 to 15 years). 2 years children were tracked and confirmed as ASD again at the age of 3.

Table 1 shows the "characteristics" of the subjects. The mean age was 7.08±2.87 years in the ASD group of children and 8.49±3.75 years in the TD group of children. In both groups, the

**Table 1. Background of TD children and ASD children subjects.**

|  |  | TD children ($n$ = 77) | ASD children ($n$ = 98) |
|---|---|---|---|
| **Age (y)** | means ± SD | 8.49±3.75 | 7.08±2.87 |
|  | median | 8 | 7 |
|  | maximum | 15 | 15 |
|  | minimum | 2 | 2 |
| **Sex** | male | 39 (50.6%) | 73 (74.5%) |
|  | female | 38 (49.4%) | 25 (25.5%) |
| **Gender comparison by age group** |  |  |  |
| 2–6 y (n) | male | 15 | 37 |
|  | female | 13 | 13 |
| 7–15 y (n) | male | 24 | 36 |
|  | female | 25 | 12 |

SD = Standard deviation; y = years.

minimum age was 2 years and the maximum age was 15 years. The proportion of males was as high as 74.5% in the ASD group, while the male to female ratio was 1:1 in the TD group. Age did not differ between the TD group and the ASD group when tested by the chi-square distribution.

Lactate and pyruvate was measured in ASD participants, and calculated the lactate/pyruvate ratio. As a result, no patients had a ratio>20, and there were no symptoms or complications suspected of mitochondrial disease.

**Indicator of oxidative stress (d-ROMs test).** For "oxidative stress evaluation", the level of excess free radicals produced in the body must be accurately measured. However, free radicals are difficult to measure in vivo because of their short lifetime and high reactivity. Thus, hydroperoxides, stable chemicals produced by oxidation of proteins, amino acids, peptides, glucosides, lipids, nucleotides and other molecules, were measured. Hydroperoxides produce free radicals (alkoxy radical, peroxy radical) in the presence of metal ions; therefore, "amount of hydroperoxides in the blood = amount of free radicals in the body." U.CARR is used for the unit, and 1U.CARR is equivalent to the $H_2O_2$ of 0.08 mg/dL. In a study by Alberti et al., They investigated the adequacy of the d-ROMs measurement and demonstrated that there is a correlation between the d-ROMs test data and the Electron Spin Resonance (ESR) assay data [21].

**Indicator of antioxidant potential (BAP test).** The BAP test measures the ability to reduce $Fe3^+$ to $Fe2^+$ and is evaluated as the ability to stop the peroxide chain reaction caused by free radicals. The principle of measurement is to first add human serum to ferric chloride ($FeCl_3$) and thiocyanate derivative (uncolored) to create a colored complex of ferric chloride with the thiocyanate derivative. Next, adding "molecule of blood plasma barrier with reducing/electron giving/antioxidant activity against ferric ions (BP (e-))", ferrous chloride ($FeCl_2$), thiocyanate derivative (uncolored) and oxidized form of BP (e-) generated. The BAP test evaluates the amount of $Fe3^+$ reduced to $Fe2^+$ in human serum by optical measurement at the bleaching level. Based on these facts, we defined "the amount of Fe ions reduced by the sample = antioxidant power". The unit is μmol / L [22].

The BAP test has been reported in metabolic syndrome [23]. In addition, various studies conducted in combination with d-ROMs have been reported to evaluate both oxidation and reduction. e.g. in conditions including diabetes, nonalcoholic steatohepatitis, epilepsy, carotid atherosclerosis [24–27].

## Measurement methods

Samples were collected from the subjects between December 2016 and September 2018. The blood samples were subjected to centrifugal separation at 1469g for 10 minutes to obtain plasma samples. The plasma levels of d-ROMs and BAP were measured using the free radical analyzer "FREE CARRIO DUO" (Diacron International, Grosscto, Italy).

## Other examination items

The subjects' scores on the Parent-interview ASD Rating Scales (PARS) [28] were examined. PARS-TR is an evaluation scale that interviews parents about the developmental and behavioral symptoms of ASD, and evaluates the presence or absence and degree. The evaluation items consist of 57 items in 6 areas, including interpersonal, communication, commitment, banding, difficulty, and irritability. The level of each item are evaluated on a three-point scale (0, 1, 2), and a higher total score suggests ASD.

To determine whether intellectual functioning has affected the results of this study, an intelligence test was conducted in the ASD participants to examine the IQ (Intelligence Quotient). And IQ<70 was defined as intellectual disability, and it was analyzed whether intellectual disability affected d-ROMs, BAP and BAP/d-ROMs.

## Statistical analysis

The relationship between gender and each test value was analyzed by the Mann–Whitney U test. The subjects were divided into TD 2–6 (TD children aged 2–6 years), ASD 2–6 (ASD children aged 2–6 years), TD 7–15 (TD children aged 7–15 years), and ASD 7–15 (ASD children aged 7–15 years), and the relationship between each age group and each test value was analyzed by the Kruskal-Wallis test. Correlation coefficients (r) and p values between the PARS scores and levels of d-ROMs and BAP were also calculated and compared among the groups. The relationship between intellectual function and d-ROMs, BAP and BAP/d-ROMs was analyzed by the Mann-Whitney U test. The statistical analysis software IBM SPSS Statistics version 21 was used.

## Ethical considerations

This study was conducted with the approval of the Ethics Committee of the Japanese Red Cross Tokushima Hinomine Rehabilitation Center for People with Disabilities. A written explanation was given to the subjects and their families, and written informed consent was obtained from the subjects, as much as possible. They were explained that they could refuse to participate or withdraw from the study at any time without any negative consequences.

## Results

Table 2 compares the gender differences among each measured value and age groups. There were no significant gender differences in the plasma levels of d-ROMs and BAP/d-ROMs ratio.

Fig 1A shows the relationship between the age groups and plasma d-ROMs levels in the TD and ASD groups of children. Among the TD children, the plasma d-ROMs levels were significantly higher in TD 2–6 than in TD 7–15 (p<0.001). ASD 7–15 had a significantly higher plasma d-ROMs level than TD 7–15 (p<0.001). However, between TD 2–6 and ASD 2–6 (p = 0.072) and between ASD 2–6 and ASD 7–15 (p = 0.171), there was no significant difference.

**Table 2. Comparison of gender differences among each measured value and age groups.**

| | Age groups (y) | Sex[a] | TD children (*n* = 77) | | ASD children (*n* = 98) | |
|---|---|---|---|---|---|---|
| | | | Mean ± SD | p value | Mean ± SD | p value |
| **d-ROMs level** | 2–6 | M | 401.4±57.1 | 0.75 | 435.2±73.6 | 0.79 |
| | | F | 409.5±63.6 | | 427.0±53.3 | |
| | 7–15 | M | 296.7±27.8 | 0.78 | 418.4±70.2 | 0.18 |
| | | F | 293.3±26.8 | | 393.4±41.4 | |
| **BAP/d-ROMs ratio** | 2–6 | M | 5.45±1.30 | 0.66 | 6.40±1.15 | 0.53 |
| | | F | 5.71±0.96 | | 6.13±0.83 | |
| | 7–15 | M | 8.11±1.06 | 0.77 | 6.23±1.18 | 0.43 |
| | | F | 7.97±0.93 | | 6.59±1.34 | |

SD = standard deviation; y = years

[a]Sex M = male, F = female.

**Fig 1B** shows the relationship between the age groups and the BAP/d-ROMs ratio in the TD and ASD groups of children. The BAP/d-ROMs ratio was significantly higher in TD 7–15 than in TD 2–6 or ASD 7–15 (p<0.001). However, between TD 2–6 and ASD 2–6 (p = 0.063) and between ASD 2–6 and ASD 7–15 (p = 0.770), there was no significant difference.

Comparison of the PARS scores between the healthy and ASD groups of children revealed that the scores were significantly higher in the ASD group of children (p<0.001). **Fig 2** shows the relationship between the PARS-TR scores and plasma d-ROMs levels in all subjects. The plasma d-ROMs levels increased as the PARS scores increased, with a significant correlation (**Fig 2A** group aged 2–6 years r = 0.343, p = 0.045; **Fig 2B** group aged 7–15 years r = 0.679, p<0.001). BAP was similarly analyzed and the results are shown in **Fig 3**.There was no significant correlation between BAP levels and PARS scores in both the 2–6 years group and the 7–15 years group. (**Fig 3A** group aged 2–6 years r = 0.211, p = 0.078; **Fig 3B** group aged 7–15 years r = 0.166, p = 0.107).

**Table 3** shows the relationship between intellectual function among ASD participants and measured values (d-ROMs, BAP and BAP/d-ROMs). When comparing the group without intellectual disability with the group with intellectual disability, no significant difference was observed in any of the measurements. (d-ROMs p = 0.821, BAP p = 0.987, BAP/d-ROMs p = 0.749).

## Discussion

d-ROMs levels (hydroperoxides levels) are substances that are converted to alkoxy and peroxy radicals in the presence of iron ions. Because it is not a reactive oxygen species or free radicals, it is a stable substance with low reactivity with oxygen. In addition, the measurement requires very small samples of blood (20μL of plasma), and numerical values are obtained within 5 minutes. Thus, this measurement is minimally invasive and provides results within a very short time [29]. Furthermore, plasma d-ROMs levels are also highly reproducible [30] and, in combination with the BAP, oxidative stress can be assessed from the aspect of both "oxidation" and "antioxidation."

Nagata et al. reported the absence of any gender differences in the plasma d-ROMs levels or BAP/d-ROMs ratio in TD adults [31]. The present study showed that there was no gender difference in children.

Comparison of the plasma d-ROMs levels between TD children and children with ASD aged 2–6 years revealed no significant differences. In both healthy and ASD groups of children

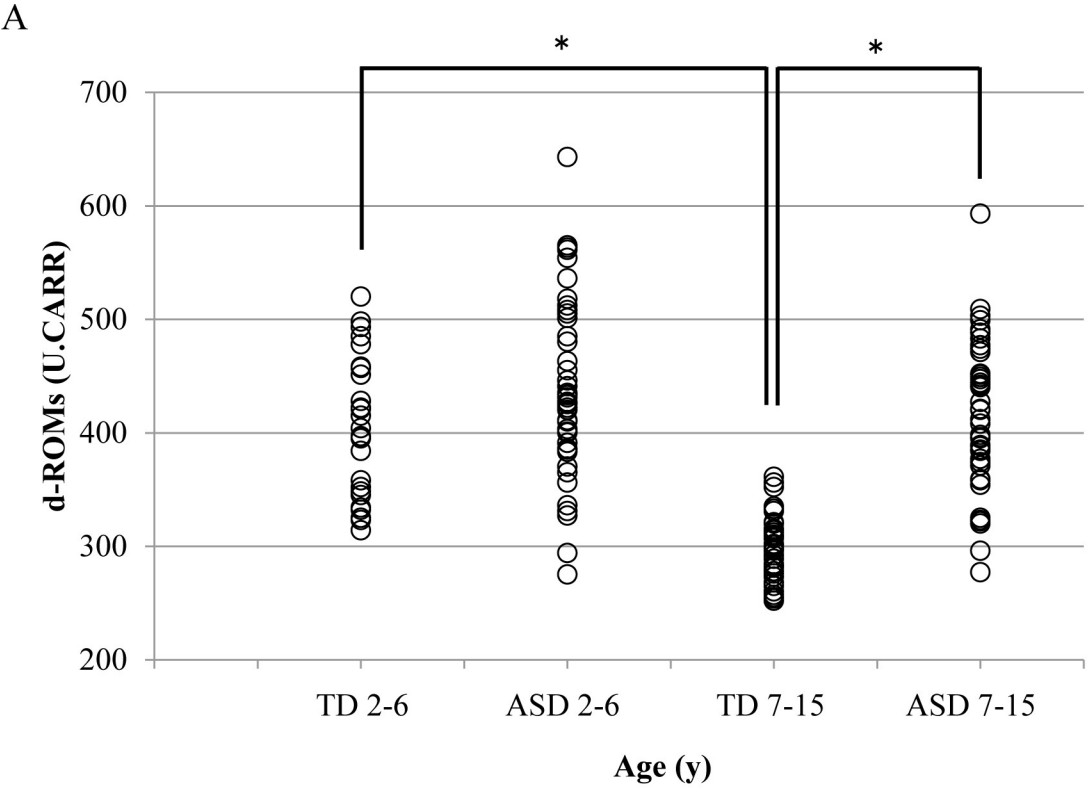

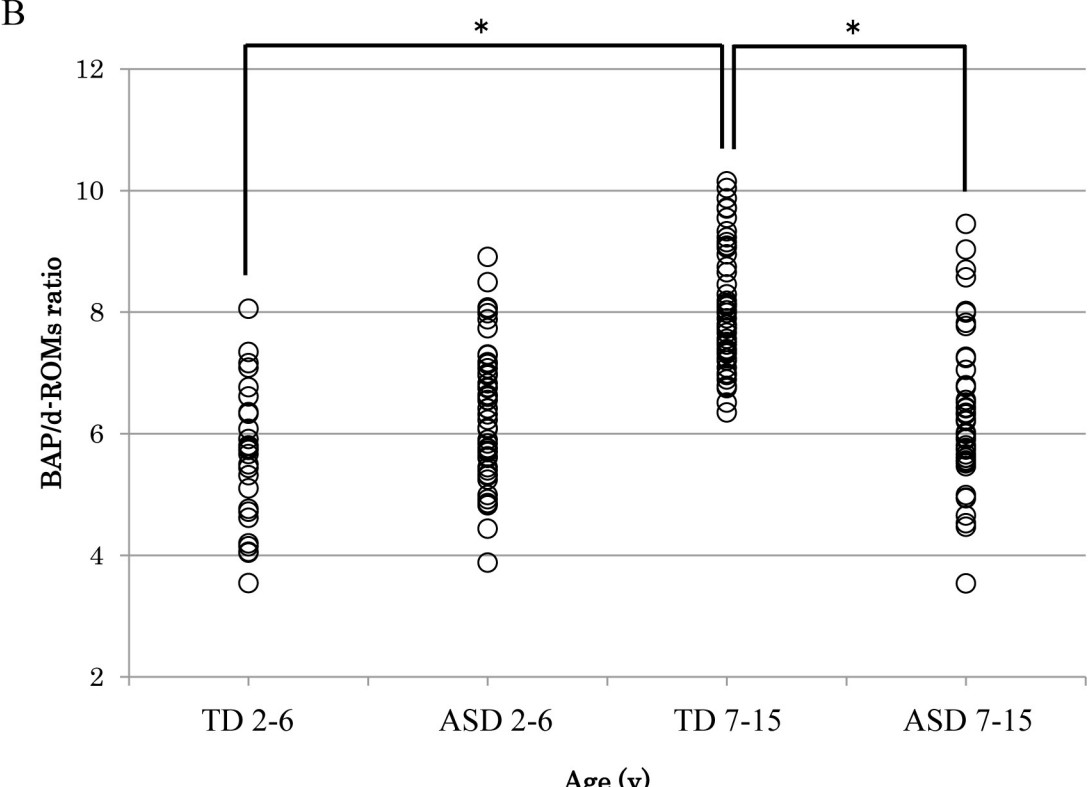

**Fig 1. Comparison of each measured level and age group.** TD 2–6 = 2–6 years TD children, ASD 2–6 = 2–6 years ASD
children, TD 7–15 = 7–15 years TD children, ASD 7–15 = 7–15 years ASD children. (A) d-ROMs levels. TD 2–6, the d-ROMs

levels were significantly higher than that in TD 7–15 (p<0.001). ASD 7–15, the d-ROMs levels were significantly higher than that in TD 7–15 (p<0.001). The mean and standard deviation of the levels in each group. 2–6 TD = 405.1±58.1, 2–6 ASD = 433.1 ±67.8, 7–15 TD = 295.0±26.8, 7–15 ASD = 412.1±64.0. (B) BAP/d-ROMs ratio. TD 7–15, the BAP/d-ROMs ratio was significantly higher than that in TD 2–6 and ASD 7–15 (p<0.001).The mean and standard deviation of the levels in each group. 2–6 TD = 5.571±1.12, 2–6 ASD = 6.337±1.06, 7–15 TD = 8.044±0.98, 7–15 ASD = 6.319±1.21.

aged 2–6 years, the higher the plasma d-ROMs level, the younger the age. Similar trends have been reported for other oxidative stress markers (acrolein-lysine, pentosidine, nitrite/nitrate and 8-hydroxy-2'-deoxyguanosine [8-OHdG]) [32, 33]. These findings are speculated to be associated with rapid synaptic pruning and regeneration in childhood [34, 35]. In addition, they may also be influenced by psychological stress due to anxiety and fear, because toddlers/ preschool age children aged 2 to 6 years have more anxiety and fear toward blood collection than school age children aged 7 to 15 years.

In contrast, comparison between TD children and children with ASD aged 7–15 years revealed significantly higher d-ROMs levels in the children with ASD. In TD children, the plasma d-ROMs levels decreased with growth and reached the reference range in TD adults (approximately < 300 U.CARR) by the age range of 7–15 years. However, in the children with ASD, the plasma d-ROMs levels failed to decrease. In addition, the BAP/d-ROMs ratio (anti- oxidant capacity) was significantly lower in the children with ASD aged 7–15 years than in those with ASD aged 2–6 years. Based on these findings, we speculate that oxidative stress may not change after growth in children with ASD, because they have a poor ability to cope with stress. Furthermore, they consider after entering school life to be more stressful due to impaired social communication and interpersonal interactions that are characteristic of ASD [36].

At the cellular level, abnormalities in mitochondria-related biomarkers, such as bisphenol A and GST, have been reported in patients with ASD [37, 38]. Free radical scavenging often depends on enzymes, and major radical-scavenging enzymes, such as glutathione peroxidase and catalase, scavenge radicals in and around the mitochondria [39], suggesting that increased plasma d-ROMs levels may be caused by mitochondrial dysfunction.

When mitochondrial dysfunction occurs due to oxidative stress, the core symptoms of ASD occur/worsen, causing stress, which induces further mitochondrial dysfunction; thus, it is speculated that a negative chain reaction is established. Therefore, periodic measurements of oxidative stress markers could be useful to assess stress in children with ASD. Rossignol DA and Frye RE focused on immune dysregulation or inflammation, oxidative stress, mitochon- drial dysfunction and environmental toxicant exposures as the four major causes of ASD, and conducted a comprehensive literature search from 1971 to 2010 investigated the relationship. As a result, it was found that several studies on lactate, pyruvate, carnitine, etc. were reported as mitochondrial-related biomarkers for ASD [40]. Simultaneous measurement of oxidative stress substances and these mitochondrial-related markers may enable more reliable assess- ment of ASD stress.

Also, the antioxidant capacity of the body is considered to be affected by the degree of oxi- dation and the total antioxidant capacity (TAC) that individuals innately possess. However, previous studies have reported only the degree of oxidation of individual indicators. Based on the results of the present study, simultaneous measurements allow the degree of oxidation and antioxidant potential to be used as useful indicators to determine the TAC in children with ASD.

In regard to the relationship between the PARS scores and plasma d-ROMs levels, there was a significant correlation in that the higher the PARS scores, the higher the plasma d- ROMs levels, suggesting an association between subjective and objective assessments. This is

A

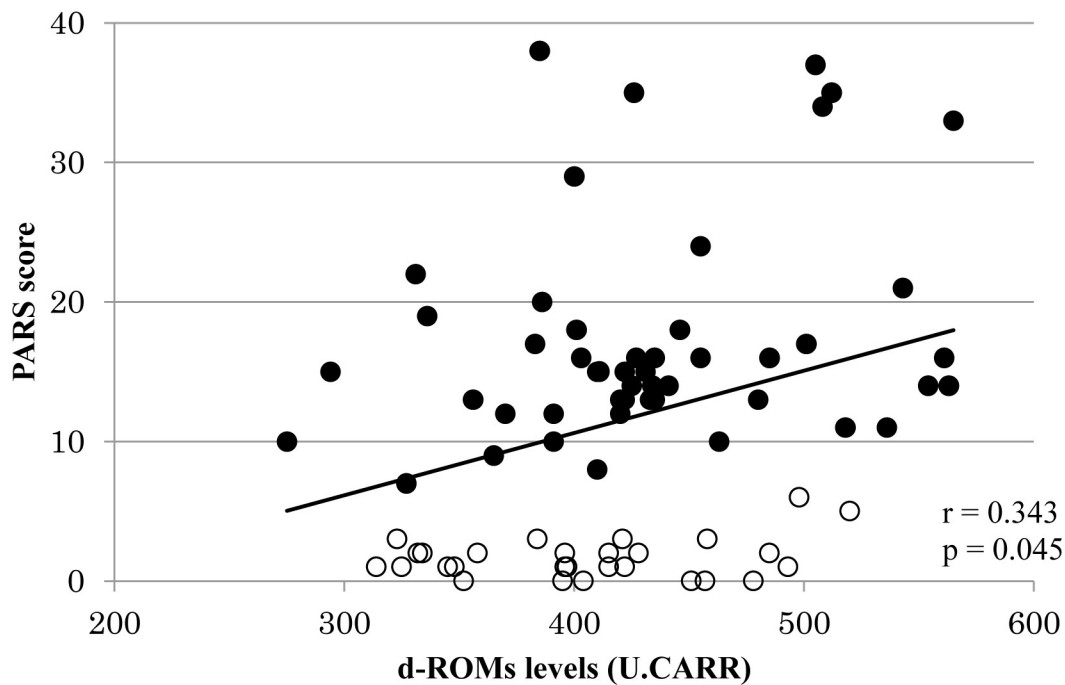

B

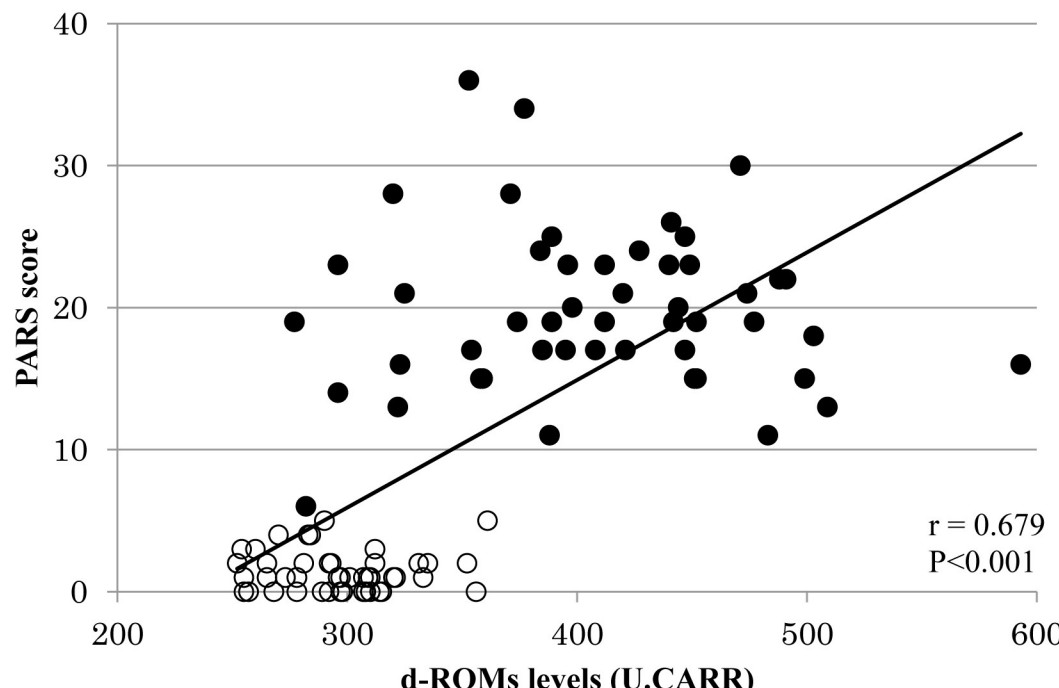

**Fig 2. Relationship between PARS scores and d-ROMs levels in all subjects.** ○ = TD children, ● = ASD children. (A) 2–6 years toddlers/preschool age children. The relationship between d-ROMs levels and PARS scores in all subjects aged 2–6 years was analyzed. As a result, a significant positive correlation was observed between the d-ROMs levels and the PARS scores. (r = 0.343, p = 0.045). (B) 7–15 years school age children. The relationship between d-ROMs levels and PARS scores in all subjects aged 7–15 years was analyzed. As a result, a significant positive correlation was observed between the d-ROMs levels and the PARS scores. (r = 0.791, p<0.001).

A

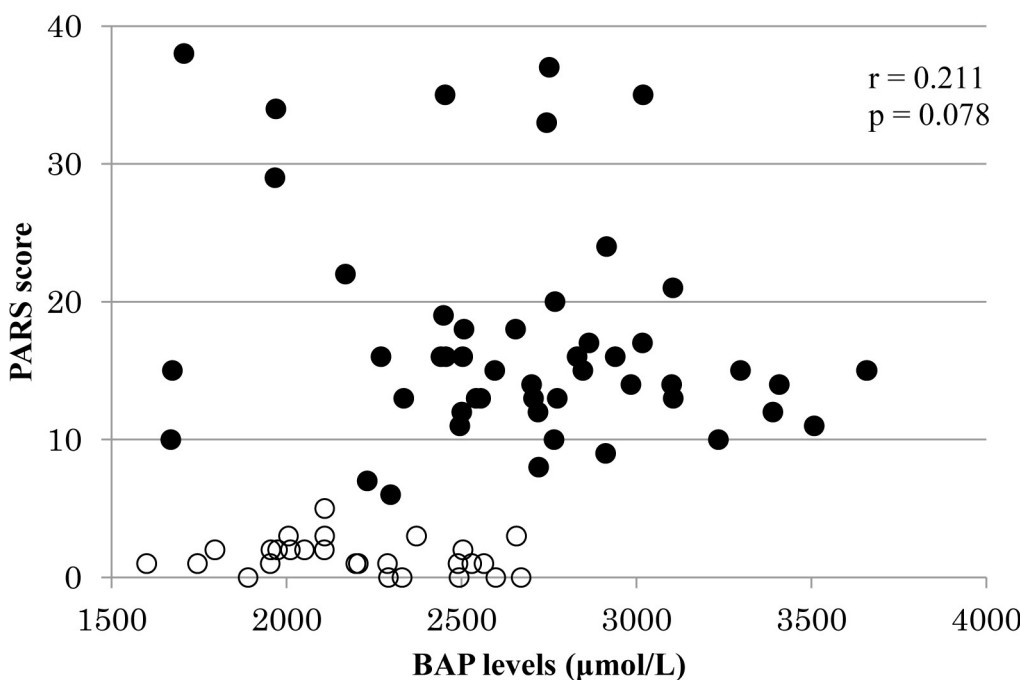

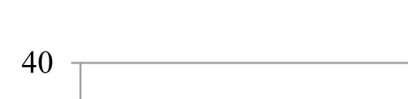

B

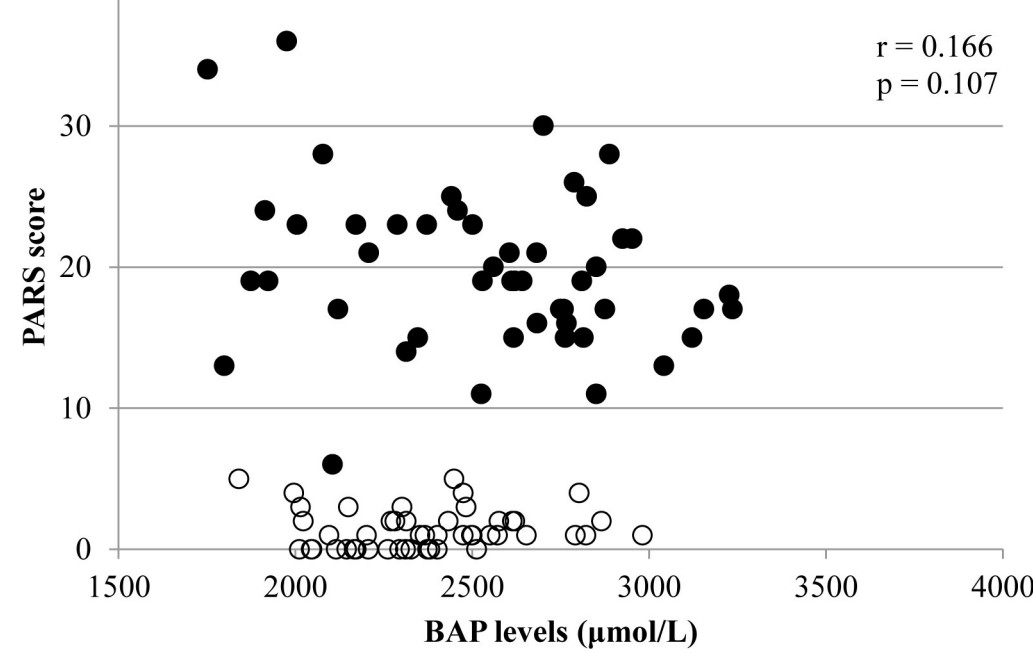

**Fig 3. Relationship between PARS scores and BAP levels in all subjects.** ○ = TD children, ● = ASD children. (A) 2–6 years toddlers/preschool age children. The relationship between BAP levels and PARS scores in all subjects aged 2–6 years

was analyzed. As a result, there was no significant correlation between BAP levels and PARS scores. (r = 0.211, p = 0.078). (B) 7–15 years school age children. The relationship between BAP levels and PARS scores in all subjects aged 7–15 years was analyzed. As a result, there was no significant correlation between BAP levels and PARS scores. (r = 0.166, p = 0.107).

**Table 3. Relationship between intelligent function and d-ROMs, BAP and BAP/d-ROMs.**

|  | IQ≧70 (*n* = 85) mean±SD | IQ<70 (*n* = 13) mean±SD | p value |
|---|---|---|---|
| d-ROMs level | 422.2±66.8 | 426.5±66.5 | 0.821 |
| BAP level | 2627.6±436.2 | 2698.0±408.4 | 0.987 |
| BAP/d-ROMs ratio | 6.316±1.162 | 6.405±0.960 | 0.749 |

IQ = intelligence quotient, SD = standard deviation

likely to increase the reliability of plasma d-ROMs as a biomarker for an intermediate phenotype of ASD. Similarly, the relationship between PARS scores and BAP levels was also investigated, but was not significantly correlated. This suggests that it is difficult to evaluate ASD only by measuring antioxidant activity using BAP.

This study had limitations. First, although various exclusion criteria were used to select the 2 groups of subjects, not all subjects may have been genuinely healthy or had definitive ASD, because the interpersonal relationships and living environment could not be examined in detail. Second, the sample size cannot be said to be sufficient. In particular, in **Fig 1**, when the d-ROMs level and the BAP/d-ROMs ratio of TD and ASD of 2–6 years old are analyzed, the d-ROMs level was p = 0.072 and the BAP/d-ROMs ratio was p = 0.063. From the results, the reason why no significant difference was found may be due to the small sample size of the TD 2–6 group. In the future, it is necessary to increase the reliability of the result on the association between ASD and oxidative stress by increasing the sample size and by combined assessment in *in vivo* and *in vitro* studies.

The results of this study may be the plasma levels of d-ROMs levels could be an objective easily measured indicator that could be used in clinical practice to assess stress and clinical state in school age children aged 7 to 15 years with ASD.

# Supporting information

**S1 Table.**
(DOCX)

# Acknowledgments

We thank the subjects for providing blood samples. We are also grateful to the nurses and clinical laboratory technicians for their assistance in blood collection and sample management.

# Author Contributions

**Conceptualization:** Shojiro Kyotani.

**Data curation:** Toshiaki Hashimoto, Yoshimi Tsuda.

**Formal analysis:** Masahito Morimoto.

**Investigation:** Masahito Morimoto, Toshiaki Hashimoto.

**Project administration:** Masahito Morimoto, Toshiaki Hashimoto, Tadanori Nakatsu, Taisuke Kitaoka.

**Supervision:** Tadanori Nakatsu, Shojiro Kyotani.

**Writing – original draft:** Masahito Morimoto, Toshiaki Hashimoto, Yoshimi Tsuda.

**Writing – review & editing:** Masahito Morimoto, Tadanori Nakatsu, Taisuke Kitaoka, Shojiro Kyotani.

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
