## [Decision Letter · Decision Letter 0]

30 Dec 2019

PONE-D-19-26530

Assessment of oxidative stress in autism spectrum disorder using reactive oxygen metabolites and biological antioxidant potential

PLOS ONE

Dear Dr Morimoto,

Thank you for submitting your manuscript to PLOS ONE. After careful consideration, we feel that it has merit but does not fully meet PLOS ONE’s publication criteria as it currently stands. Therefore, we invite you to submit a revised version of the manuscript that addresses the points raised during the review process.

We would appreciate receiving your revised manuscript by Feb 10 2020 11:59PM. To enhance the reproducibility of your results, we recommend that if applicable you deposit your laboratory protocols in protocols.io, where a protocol can be assigned its own identifier (DOI) such that it can be cited independently in the future. For instructions see: http://journals.plos.org/plosone/s/submission-guidelines#loc-laboratory-protocols

We look forward to receiving your revised manuscript.

Kind regards,

Ece Uzun, PhD

Academic Editor

PLOS ONE

Journal Requirements:

Additional Editor Comments (if provided):

In addition to the reviewers' comments, please include the following points in your reviews:

Introduction: Line 44 You talk about secondary symptoms in autism, but these are co-morbidities and a certain percentage of patients have those. Please clarify this and add what percentage of patients have insomnia, depression and anxiety.

Line 49: Please reference the last sentence in paragraph 1.

Line 57: Please explain in detail why the markers can not be measured rapidly? Please clarify what you mean from rapidly? How long does it take to run the test and what is the benefit of measuring it fast vs slow?

Line 63:Please explain the biological relevance of d-ROMS and BAP in autism.

Methods: Is "Measurement items" a sub-title? Please exclude this as this section is part of methods. Please follow the author guidelines fro proper sub-section titles.

Line 92: What does ESR stand for?

Figures: Please exclude "This the Fig 1 Title" or "This is the Fig 1 legend" statements from all figures. Please follow the author guidelines for proper figure title, legend and captions. Please explain the results observed in a specific figure in detail in the figure captions.

Results: First paragraph should be moved to Methods section as it is describing the cohort. I would recommend to start the results section with a brief introduction of what was done in the study following with the description of results.

Reviewers' comments:

Reviewer's Responses to Questions

**Comments to the Author**

1. Is the manuscript technically sound, and do the data support the conclusions?

Reviewer #1: Partly

Reviewer #2: Partly

2. Has the statistical analysis been performed appropriately and rigorously? 

Reviewer #1: I Don't Know

Reviewer #2: Yes

3. Have the authors made all data underlying the findings in their manuscript fully available?

Reviewer #1: Yes

Reviewer #2: Yes

4. Is the manuscript presented in an intelligible fashion and written in standard English?

Reviewer #1: Yes

Reviewer #2: Yes

5. Review Comments to the Author

Reviewer #1: Comments to the Authors:

Autism spectrum disorder (ASD) is a group of complicated neurodevelopmental disorders characterized by difficulties in social communication and interactions with restricted repetitive behaviors or interests. It causes by an interaction between genetic vulnerability and environmental factors. In order to better understand roots of ASD for diagnosis and treatment, efforts to identify reliable biomarkers are growing. This manuscript investigated whether d-ROMs and BAP/d-ROMs ratio could be objective indicators to assess oxidative stress in untreated ASD children. To address this, the authors measured the plasma d-ROMs levels and BAP/d-ROMs ratio in ASD and typical development (TD) children in two different age groups (2-6y and 7-15y) simultaneously. They found that the levels of d-ROMs were significantly higher in the ASD (7-15 years) than in TD (7-15 years). They also found that Parent-interview ASD Rating Scales (PARS) scores were significantly higher in the ASD and were significantly correlated with d-ROMs levels. This work is informative; however, I have a number of observations and comments:

1. The major issue of this study is the small sample size. This study included 77 TD children and 98 children with untreated ASD. Based on their age, they were separated into two age groups which made each group less number. Especially the 2-6Y TD group. They didn’t find significant difference in plasma d-ROM level and BAP/d-ROMs ratio between ASD and TD in 2-6Y group. Could that be because of the small sample size?

2. The authors claimed that PARS scores were significantly correlated with the plasma d-ROMs levels. However, for 2-6y old, the correlation is weak (r=0.343). How they calculated p values should be clarified and discussed.

3. The authors thought that increased plasma d-ROMs levels may be caused by mitochondrial dysfunction. Is it possible to test some mitochondria-related biomarkers in these samples?

4. For Fig1: it is confusing to label X axis with A, B, C and D. Labeling each group with the age and TD or ASD will be better. Also, please put mean and error bar in the data too.

5. For Fig2: please add figure legend and r value in the figure.

Reviewer #2: PONE-D-19-26530

The authors have explored oxidative stress in ASD with an approach that appears to bring new elements. It is of interest, but there are some issues regarding the communication of the findings.

A minor point in the abstract. The last line ‘indicator’ should be ‘indicators’.

In the introduction-should point out that ‘suicidal ideation’ is based on rather recent reports, and was not previously known. Also, the last sentence of the first paragraph needs to be fully referenced. Also ‘various biomarkers have been explored’ only cites a paper examining clinical features of children who pass 18-month screening. Not sure how this is related. There are MANY efforts at biomarker exploration that deserve mention, and none are cited here. For example, Amaral’s group looking for metabolomic markers, and the work of Hicks and Middleton’s group looking at RNA markers, both of which have resulted in the formation of companies for ASD biomarkers would seem a starting point.

Methods- it is remarkable that they have found such a sample of ASD patients with no medications, minimizing risk of confounds from drug effects, but should mention in the Discussion the potential limitation of exclusion of patients that were more severe and needed medications. ‘Since age and living environment may effect stress’- should briefly explain the rationale for this initially in paragraph 2 when it is first mentioned. The understanding of this doesn’t become clear until late in the Results. Next section, change ‘Hydeoperoxide, which was’ to ‘Hydroperoxides, which are’, and expand in the background on the ‘produced by oxidation of proteins, amino acids, peptides, glucosides, lipids, nucleotides, and other molecules’- please explain this further so that the reader gets a better understanding of why this is being explored. Later ‘ferric to ferrous’ and ‘reduce ferric’- please make a more complete description of these instead of the shorthand notation here, and give more on how this indicator is used elsewhere to help the reader understand why it is being used here. Briefly describe the PARS. Finally, in the Analysis, was the correlation between PARS and BAP also assessed?

Results- were the ages statistically different between groups? Was there any indicator of intellectual functioning, to show that this is specific to ASD rather than intellectual disability? Also, was gender tracked in the statistics since the groups differed in gender balance? Also, please point out in the text the pertinent negative findings, such as the lack of difference between TD and ASD for ages 2-6 for d-ROMs. Otherwise the reader might not realize this was a negative finding, causing confusion later in reading the paper. Fig. 1 title change ‘levels’ to ‘level’.

Discussion- it is stated that ‘plasma d-ROMs levels are highly stable’ but the reason offered is that they are quantitatively measured. Is that why they are stable? Please clarify further the reason why they are stable. Please provide a reference for ‘speculated to be related to rapid cellular apoptosis and regeneration in childhood’, and later, references for ‘children with ASD have high degrees of psychological stress’ and ‘the fact that these children are always exposed to the stress of group living after entering school’. Later, in the discussion on ‘the antioxidant capacity of the body’, should discuss the findings on mitochondrial makers in autism (such as the paper by Rossignol and Frye, Mol Psychiatry 2012;17:290-314), as this is highly relevant. In the paragraph discussing the relation between PARS and d-ROMs, should again note if there was no such relationship for BAP. At the end, this finding is FAR too premature to make the recommendation that this is ‘an objective easily measured indicator that could be used in clinical practice to assess stress’, and it is not reasonable to make a claim such as ‘useful for assessing the clinical severity of ASD’. The actual clinical severity of ASD is inherently more reliable as an indicator of severity-and this marker would not have an impact on this measure.

6. PLOS authors have the option to publish the peer review history of their article (what does this mean?). If published, this will include your full peer review and any attached files.

Reviewer #1: No

Reviewer #2: Yes: David Q. Beversdorf, MD

---

## [Author Response · Author response to Decision Letter 0]

12 Feb 2020

3 February, 2020

Dear Reviewer #1

Thank you for inviting us to submit a revised draft of our manuscript entitled, “Assessment of Oxidative Stress in Autism Spectrum Disorder Using Reactive Oxygen Metabolites and Biological Antioxidant Potential” to PLOS ONE. We also appreciate the time and effort you and each of the reviewers have dedicated to providing insightful feedback on ways to strengthen our paper. Thus, it is with great pleasure that we resubmit our article for further consideration. We have incorporated changes that reflect the detailed suggestions you have graciously provided. We also hope that our edits and the responses we provide below satisfactorily address all the issues and concerns you and the reviewers have noted.

Sincerely yours, 

Masahito Morimoto

Corresponding author

Our response to the reviewer comment is as follow:

Comment 1: 

The major issue of this study is the small sample size. This study included 77 TD children and 98 children with untreated ASD. Based on their age, they were separated into two age groups which made each group less number. Especially the 2-6Y TD group. They didn’t find significant difference in plasma d-ROM level and BAP/d-ROMs ratio between ASD and TD in 2-6Y group. Could that be because of the small sample size?

Response:

We wish to thank the Reviewer for this comment.

Small sample sizes are also described in the study limits in the discussion. We have shown in previous studies that oxidative stress changes with age. So we had to split it into two age groups. Therefore, especially the TD 2-6 Y group has a small sample size.

Analyzing the d-ROMs levels and BAP/d-ROMs ratio for TD and ASD of 2-6 years, the d-ROMs levels were p=0.072 and the BAP/d-ROMs ratio was p=0.063 (Man Whitney U test). From the results, the reason why no significant difference was found may be due to the small sample size of the TD 2-6 Y group.

Add these contents to the discussion of the manuscript.

Comment 2: 

The authors claimed that PARS scores were significantly correlated with the plasma d-ROMs levels. However, for 2-6y old, the correlation is weak (r=0.343). How they calculated p values should be clarified and discussed.

Response:

Thank you very much for your point. When statistical analysis was performed again, it was found that there was an error in the p-value. It was changed from p = 0.004 to p = 0.045. 

We really appreciate point out important errors.

Comment 3: 

The authors thought that increased plasma d-ROMs levels may be caused by mitochondrial dysfunction. Is it possible to test some mitochondria-related biomarkers in these samples?

Response:

We appreciate the Reviewer's comment on this point. The subject's ASD measured lactate and pyruvate as mitochondrial-related biomarkers. I added this to the "Methods" section as follows.

ASD measured lactate and pyruvate as mitochondrial biomarkers and calculated the lactate/pyruvate ratio. As a result, no patients had a ratio>20, and there were no symptoms or complications suspected of mitochondrial disease.

Comment 4: 

For Fig1: it is confusing to label X axis with A, B, C and D. Labeling each group with the age and TD or ASD will be better. Also, please put mean and error bar in the data too.

Response:

We wish to thank the Reviewer for this comment.

As you pointed out, the labels on the X axis are difficult to understand, so they have been revised as follows. Group A was changed to "TD 2-6", Group B was changed to "ASD 2-6", Group C was changed to "TD 7-15" and Group D was changed to "ASD 7-15".

We were unable to add the mean and error bars to the scatter plot (Fig 1) created by special processing. So, as an alternative, we added the mean and standard deviation to the figure legend.

Comment 5: 

For Fig2: please add figure legend and r value in the figure.

Response:

As you pointed out, the figure legend and r value are added to Fig2.

Again, thank you for giving us the opportunity to strengthen our manuscript with your valuable comments and queries. We have worked hard to incorporate your feedback and hope that these revisions persuade you to accept our submission.

Masahito Morimoto (Corresponding Author)

Japanese Red Cross Tokushima Hinomine Rehabilitation Center for People with Disabilities 

Address: 4-1Shinbiraki, chuden-cho, Komathushima-shi, Tokushima, 773-0015 Japan 

Phone : +81-885-32-0903 Fax : +81-885-33-3037

E-mail address: morimoto@hinomine-mrc.jp

3 February, 2020

Dear Prof. David Q. Beversdorf, MD (Reviewer #2)

Thank you for inviting us to submit a revised draft of our manuscript entitled, “Assessment of Oxidative Stress in Autism Spectrum Disorder Using Reactive Oxygen Metabolites and Biological Antioxidant Potential” to PLOS ONE. We also appreciate the time and effort you and each of the reviewers have dedicated to providing insightful feedback on ways to strengthen our paper. Thus, it is with great pleasure that we resubmit our article for further consideration. We have incorporated changes that reflect the detailed suggestions you have graciously provided. We also hope that our edits and the responses we provide below satisfactorily address all the issues and concerns you and the reviewers have noted.

Sincerely yours, 

Masahito Morimoto

Corresponding author

Our response to the reviewer comment is as follow:

Comment 1: 

The last line ‘indicator’ should be ‘indicators’.

Response:

Revised as pointed out.

Comment 2: 

Introduction- In the introduction-should point out that ‘suicidal ideation’ is based on rather recent reports, and was not previously known.

Response:

We strongly appreciate the Reviewer's comment on this point. As you pointed out, "suicidal ideation" has been clarified in a recent report and will be removed from the text.

Comment 3: 

Introduction- The last sentence of the first paragraph needs to be fully referenced. 

Also ‘various biomarkers have been explored’ only cites a paper examining clinical features of children who pass 18-month screening. Not sure how this is related. There are MANY efforts at biomarker exploration that deserve mention, and none are cited here.

Response:

We thank the Reviewer for this pertinent comment.

Yes, I added references. As you pointed out, reference number 5 was not a valid reference. Therefore, we removed this reference.

Comment 4: 

Methods- it is remarkable that they have found such a sample of ASD patients with no medications, minimizing risk of confounds from drug effects, but should mention in the Discussion the potential limitation of exclusion of patients that were more severe and needed medications. ‘Since age and living environment may effect stress’ - should briefly explain the rationale for this initially in paragraph 2 when it is first mentioned. The understanding of this doesn’t become clear until late in the Results.

Response:

We thank the Reviewer for this insightful comment.

Regarding 'Since age and living environment may effect stress', the following content was added to paragraph 2 and explained.

In our previous study, younger children had higher oxidative stress levels. And it has been reported that the stress level of children differs due to changes in the environment between preschool and school life.

Comment 5: 

Methods- Next section, change ‘Hydeoperoxide, which was’ to ‘Hydroperoxides, which are’, and expand in the background on the ‘produced by oxidation of proteins, amino acids, peptides, glucosides, lipids, nucleotides, and other molecules’- please explain this further so that the reader gets a better understanding of why this is being explored.

Response:

We appreciate the Reviewer's comment on this point.

‘Hydeoperoxide, was changed to ‘Hydeoperoxides,.

The text has been revised as follows to help readers understand why hydroperoxides were measured in this study.

For "oxidative stress evaluation", the level of excess free radicals produced in the body must be accurately measured. However, free radicals are difficult to measure in vivo because of their short lifetime and high reactivity. Thus, hydroperoxides, stable chemicals produced by oxidation of proteins, amino acids, peptides, glucosides, lipids, nucleotides and other molecules, were measured. Hydroperoxides produce free radicals (alkoxy radical, peroxy radical) in the presence of metal ions; therefore, “amount of hydroperoxides in the blood = amount of free radicals in the body.”

Comment 6: 

Methods- Later ‘ferric to ferrous’ and ‘reduce ferric’- please make a more complete description of these instead of the shorthand notation here, and give more on how this indicator is used elsewhere to help the reader understand why it is being used here.

Response:

We strongly appreciate the Reviewer's comment on this point.

The BAP test has been revised with detailed explanations. Also added with references to how BAP is used in other studies. The details of the revision are described below.

The BAP test measures the ability to reduce Fe3+ to Fe2+ and is evaluated as the ability to stop the peroxide chain reaction caused by free radicals. The principle of measurement is to first add human serum to ferric chloride (FeCl3) and thiocyanate derivative (uncolored) to create a colored complex of ferric chloride with the thiocyanate derivative. Next, adding “molecule of blood plasma barrier with reducing / electron giving/antioxidant activity against ferric ions (BP (e-))”, ferrous chloride (FeCl2), thiocyanate derivative (uncolored) and oxidized form of BP (e-) generates. The BAP test evaluates the amount of Fe3+ reduced to Fe2+ in human serum by optical measurement at the bleaching level. Based on these facts, we defined “the amount of Fe ions reduced by the sample = antioxidant power”. The unit is μmol / L [Ref].

The BAP test has a report on metabolic syndrome [Ref]. In addition, various studies conducted in combination with d-ROMs have been reported to evaluate both oxidation and reduction. e.g. diabetes, nonalcoholic steatohepatitis, epilepsy, carotid atherosclerosis [Ref].

Comment 7: 

Methods- Briefly describe the PARS.

Response:

The following explanation has been added.

PARS-TR is an evaluation scale that interviews parents about the developmental and behavioral symptoms of ASD, and evaluates the presence or absence and degree. The evaluation items consist of 57 items in 6 areas, including interpersonal, communication, commitment, banding, difficulty, and irritability. The levels of each items are evaluated on a three-point scale (0, 1, 2), and a higher total score strongly suggests ASD.

Comment 8: 

Methods- In the Analysis, was the correlation between PARS and BAP also assessed?

Response:

We appreciate the Reviewer's comment on this point. We investigated and analyzed the correlation between PARS and BAP. PARS and BAP were not significantly correlated in both the 2-6 and 7-15 years groups. This result is shown in Fig. 3. This content was added to "Methods", "Results" and "Discussion".

Comment 9: 

Results- Were the ages statistically different between groups?

Response:

We appreciate the Reviewer's comment on this point. Age did not differ between the TD group and the ASD group when tested by the chi-square distribution. Analysis results were 2-6 years: p=0.276, 7-15 years: p=0.425. This has been added to the subjects section of the methods. 

Comment 10: 

Results- Was there any indicator of intellectual functioning, to show that this is specific to ASD rather than intellectual disability?

Response:

We thank the Reviewer for this insightful comment. ASD had been tested for intelligence. Defining IQ <70 as intellectual disability and statistically analyzing whether intellectual functions affected d-ROMs, BAP and BAP / d-ROMs, there was no significant difference. Therefore, this study does not appear to be affected by intellectual disability. This has been added to the Methods and Results section.

Comment 11: 

Results- Was gender tracked in the statistics since the groups differed in gender balance?

Response:

Table 2 shows the gender, measurements, and analysis results for each group. Statistical analysis revealed no gender differences in each of the groups.

Comment 12: 

Results- Please point out in the text the pertinent negative findings, such as the lack of difference between TD and ASD for ages 2-6 for d-ROMs.

Response:

We thank the Reviewer for this pertinent comment. Added the results of this study, especially the negative findings in Fig 1 and Fig 2.

Comment 13: 

Results- Fig. 1 title change ‘levels’ to ‘level’.

Response:

The title of Fig.1 has been changed from "levels" to "level".

Comment 14: 

Discussion- it is stated that ‘plasma d-ROMs levels are highly stable’ but the reason offered is that they are quantitatively measured. Is that why they are stable? Please clarify further the reason why they are stable.

Response:

We appreciate the Reviewer's comment on this point. The description of the reason why the d-ROMs level is stable has been changed and the reason why the level is stable has been added as follows.

d-ROMs level (hydroperoxides) are substances that are converted to alkoxy and peroxy radicals in the presence of iron ions. Because it is not a reactive oxygen species or free radical, it is a stable substance with low reactivity with oxygen.

Comment 15: 

Discussion- Please provide a reference for ‘speculated to be related to rapid cellular apoptosis and regeneration in childhood’, and later, references for ‘children with ASD have high degrees of psychological stress’ and ‘the fact that these children are always exposed to the stress of group living after entering school’.

Response:

We thank the Reviewer for this pertinent comment. As you pointed out, we provided references for ' These findings are speculated to be associated with rapid synaptic pruning and regeneration in childhood'. And the text was changed to "These findings are speculated to be associated with rapid synaptic pruning and regeneration in childhood."

‘Children with ASD have high degrees of psychological stress' and ‘the fact that these children are always exposed to the stress of group living after entering school' was changed as follows and added reference.

‘They consider after entering school life to be more stressful due to impaired social communication and interpersonal interactions that are characteristic of ASD'. 

Comment 16: 

Discussion- In the discussion on ‘the antioxidant capacity of the body’, should discuss the findings on mitochondrial makers in autism (such as the paper by Rossignol and Frye, Mol Psychiatry 2012;17:290-314), as this is highly relevant.

Response:

We strongly appreciate the Reviewer's comment and suggestion. We have read the review of Rossignol DA and Frye RE. The findings of oxidative stress and mitochondrial-related markers in ASD were discussed with the results of this study and added to the Discussion.

Comment 17: 

Discussion- In the paragraph discussing the relation between PARS and d-ROMs, should again note if there was no such relationship for BAP. 

Response:

In connection with the response to comment 8, added the following to the paragraph pointed to you.

Similarly, the relationship between PARS scores and BAP levels was also investigated, but was not significantly correlated. This suggests that it is difficult to evaluate ASD only by measuring antioxidant activity using BAP.

Comment 18: 

Discussion- This finding is FAR too premature to make the recommendation that this is ‘an objective easily measured indicator that could be used in clinical practice to assess stress’, and it is not reasonable to make a claim such as ‘useful for assessing the clinical severity of ASD’. The actual clinical severity of ASD is inherently more reliable as an indicator of severity-and this marker would not have an impact on this measure.

Response:

We thank the Reviewer for this pertinent comment. As you pointed out, the conclusions are overstatement that cannot be proved by the results of this study. Therefore, the last sentence, “In addition, the plasma d-ROMs levels were correlated with the PARS scores, suggesting that measurement of the plasma d-ROMs levels could also be useful for assessing the clinical severity of ASD.” was deleted.

Again, thank you for giving us the opportunity to strengthen our manuscript with your valuable comments and queries. We have worked hard to incorporate your feedback and hope that these revisions persuade you to accept our submission.

Masahito Morimoto (Corresponding Author)

Japanese Red Cross Tokushima Hinomine Rehabilitation Center for People with Disabilities 

Address: 4-1Shinbiraki, chuden-cho, Komathushima-shi, Tokushima, 773-0015 Japan 

Phone : +81-885-32-0903 Fax : +81-885-33-3037

E-mail address: morimoto@hinomine-mrc.jp

---

## [Decision Letter · Decision Letter 1]

13 Apr 2020

PONE-D-19-26530R1

Assessment of oxidative stress in autism spectrum disorder using reactive oxygen metabolites and biological antioxidant potential

PLOS ONE

Dear Dr. Morimoto,

Thank you for submitting your manuscript to PLOS ONE. I apologize for the delay in our decision. After careful consideration, we feel that it has merit but does not fully meet PLOS ONE’s publication criteria as it currently stands. Therefore, we invite you to submit a revised version of the manuscript that addresses the points raised during the review process.

Please review the comments from the Reviewers. Please proof-read the text and change some of the poor wording commented by Reviewer 2.

We would appreciate receiving your revised manuscript by April 27, 2020. To enhance the reproducibility of your results, we recommend that if applicable you deposit your laboratory protocols in protocols.io, where a protocol can be assigned its own identifier (DOI) such that it can be cited independently in the future. For instructions see: http://journals.plos.org/plosone/s/submission-guidelines#loc-laboratory-protocols

We look forward to receiving your revised manuscript.

Kind regards,

Ece Uzun, PhD

Academic Editor

PLOS ONE

Reviewers' comments:

Reviewer's Responses to Questions

**Comments to the Author**

1. If the authors have adequately addressed your comments raised in a previous round of review and you feel that this manuscript is now acceptable for publication, you may indicate that here to bypass the “Comments to the Author” section, enter your conflict of interest statement in the “Confidential to Editor” section, and submit your "Accept" recommendation.

Reviewer #1: All comments have been addressed

Reviewer #2: (No Response)

2. Is the manuscript technically sound, and do the data support the conclusions?

Reviewer #1: Yes

Reviewer #2: Yes

3. Has the statistical analysis been performed appropriately and rigorously? 

Reviewer #1: Yes

Reviewer #2: Yes

4. Have the authors made all data underlying the findings in their manuscript fully available?

Reviewer #1: Yes

Reviewer #2: Yes

5. Is the manuscript presented in an intelligible fashion and written in standard English?

Reviewer #1: Yes

Reviewer #2: Yes

6. Review Comments to the Author

Reviewer #1: (No Response)

Reviewer #2: PONE-D-19-26530 R1

The authors have explored oxidative stress in ASD with an approach that appears to bring new elements. It is of interest, and the authors have been responsive to comments. Only a few issues remain with the new text.

In the introduction-line 61 -wording is clumsy, change ‘that tests that’ to ‘that tests which’- less redundant. Also, change ‘illnesses’ to ‘conditions’- as ASD is not really an ‘illness’. Also be consistent on capitalization for ‘Fenton’.

Methods- Wording is bad on lined 103-104 ‘Analysis results were 2-6 years p = 0.276, 7-15 years p = 0.425. ASD measured lactate and pyruvate’ I don’t know what the first sentence means at all, and how did ‘ASD’ measure lactate and pyruvate? Do the authors mean ‘Lactate and pyruvate was measured in ASD participants’? Please clarify. Line 134- change ‘generates’ to ‘is generated’. Line 138- change ‘has a report on metabolic syndrome’ to ‘has been reported in metabolic syndrome’ and insert ‘in conditions including’ before ‘diabetes’ on line 140. Lines 159-160, change ‘items’ to ‘item’ and don’t need ‘strongly’. Line 162, change ‘ASD conducted an intelligence test’ to ‘an intelligence test was conducted in the ASD participants’.

Results- Table 3 (line 286) change ‘intellectual functions of ASD’ to ‘intellectual function among ASD participants’.

Discussion- at the end- still over stated- this finding is FAR too premature to make the recommendation that this is ‘an objective easily measured indicator that could be used in clinical practice to assess stress and clinical state’. It may certainly be worthy of further exploration, but this study is not sufficient to make such a clinical practice recommendation at this time.

7. PLOS authors have the option to publish the peer review history of their article (what does this mean?). If published, this will include your full peer review and any attached files.

Reviewer #1: No

Reviewer #2: Yes: David Q. Beversdorf, MD

---

## [Author Response · Author response to Decision Letter 1]

17 Apr 2020

April 17, 2020

Dear Reviewer #2

We thank the reviewer for fruitful suggestions, especially for suggesting the better terms and sentences. We have revised the manuscript PONE-D-19-26530R1 “Assessment of oxidative stress in autism spectrum disorder using reactive oxygen metabolites and biological antioxidant potential” on the basis of the reviewer's comments. 

We look forward to a publication of our manuscript in PLOS ONE.

Sincerely yours, 

Masahito Morimoto

Corresponding author

Our response to the reviewer comment is as follow:

Comment 1: 

In the introduction-line 61 -wording is clumsy, change‘that tests that’ to ‘that tests which’- less redundant. 

Response:

We corrected as you pointed out.

Comment 2: 

Also, change ‘illnesses’ to ‘conditions’- as ASD is not really an ‘illness’.

Response:

We corrected as you pointed out.

Comment 3: 

Also be consistent on capitalization for ‘Fenton’.

Response:

We become consistent on capitalization for ‘Fenton’.

Comment 4: 

Wording is bad on lined 103-104 ‘Analysis results were 2-6 years p = 0.276, 7-15 years p = 0.425.

Response:

We deleted the sentence ‘Analysis results were 2-6 years p = 0.276, 7-15 years p = 0.425.’, because it was irrelevant to the explanation in Table 1.

Comment 5: 

ASD measured lactate and pyruvate’ I don’t know what the first sentence means at all, and how did ‘ASD’ measure lactate and pyruvate? Do the authors mean ‘Lactate and pyruvate was measured in ASD participants’? Please clarify.

Response:

We appreciate the Reviewer's comment on this point.

I changed the text as follows.

`Lactate and pyruvate was measured in ASD participants, and calculated the lactate/pyruvate ratio. As a result, no patients had a ratio>20, and there were no symptoms or complications suspected of mitochondrial disease.'

And since it is not related to Table 1, we wrote it in another paragraph.

Comment 6: 

Line 134- change ‘generates’ to ‘is generated’.

Response:

We corrected as you pointed out.

Comment 7: 

Line 138- change ‘has a report on metabolic syndrome’ to ‘has been reported in metabolic syndrome’ and insert ‘in conditions including’ before ‘diabetes’ on line 140.

Response:

We corrected as you pointed out.

Comment 8: 

Lines 159-160, change ‘items’ to ‘item’ and don’t need ‘strongly’.

Response:

We corrected as you pointed out.

Comment 9: 

Line 162, change ‘ASD conducted an intelligence test’ to ‘an intelligence test was conducted in the ASD participants’.

Response:

We corrected as you pointed out.

Comment 10: 

Results- Table 3 (line 286) change ‘intellectual functions of ASD’ to ‘intellectual function among ASD participants’.

Response:

We corrected as you pointed out.

Comment 11: 

Discussion- at the end- still over stated- this finding is FAR too premature to make the recommendation that this is ‘an objective easily measured indicator that could be used in clinical practice to assess stress and clinical state’. It may certainly be worthy of further exploration, but this study is not sufficient to make such a clinical practice recommendation at this time.

Response:

We appreciate the Reviewer's comment on this point.

As you pointed out, the results of this study do not currently have any solid evidence to use in clinical practice. Therefore, we changed "suggest" to "may be" to soften the expression.

Again, thank you for giving us the opportunity to strengthen our manuscript with your valuable comments and queries. We have worked hard to incorporate your feedback and hope that these revisions persuade you to accept our submission.

Masahito Morimoto (Corresponding Author)

Japanese Red Cross Tokushima Hinomine Rehabilitation Center for People with Disabilities 

Address: 4-1Shinbiraki, chuden-cho, Komathushima-shi, Tokushima, 773-0015 Japan 

Phone : +81-885-32-0903 Fax : +81-885-33-3037

E-mail address: morimoto@hinomine-mrc.jp

---

## [Editor Report · Decision Letter 2]

8 May 2020

Assessment of oxidative stress in autism spectrum disorder using reactive oxygen metabolites and biological antioxidant potential

PONE-D-19-26530R2

Dear Dr. Morimoto,

We are pleased to inform you that your manuscript has been judged scientifically suitable for publication and will be formally accepted for publication once it complies with all outstanding technical requirements.

With kind regards,

Ece Uzun, PhD

Academic Editor

PLOS ONE
---

## [Editor Report · Acceptance letter]

12 May 2020

PONE-D-19-26530R2 

Assessment of oxidative stress in autism spectrum disorder using reactive oxygen metabolites and biological antioxidant potential 

Dear Dr. Morimoto:

I am pleased to inform you that your manuscript has been deemed suitable for publication in PLOS ONE. Congratulations! Your manuscript is now with our production department. 

With kind regards,

on behalf of

Dr. Ece Uzun 

Academic Editor

PLOS ONE